# Full Digital Workflow for the Treatment of an Edentulous Patient with Guided Surgery, Immediate Loading and 3D-Printed Hybrid Prosthesis: The BARI Technique 2.0. A Case Report

**DOI:** 10.3390/ijerph16245160

**Published:** 2019-12-17

**Authors:** Pietro Venezia, Ferruccio Torsello, Vincenzo Santomauro, Vittorio Dibello, Raffaele Cavalcanti

**Affiliations:** 1Department of Prosthodontics, Section of Dentistry, University of Catania, 95124 Catania, Italy; 2Department of Periodontics and Prosthodontics Eastman Dental Hospital, 00161 Rome, Italy; ferruccio@torsello.it; 3Private Practice, 84091 Battipaglia (SA), Italy; santomaurovincenzo@virgilio.it; 4Interdisciplinary Department of Medicine, University Aldo Moro, 70121 Bari, Italy; vittoriodibello1@gmail.com; 5Section of Periodontology, Department of Oral and Maxillo-Facial Sciences, “Sapienza” University, 00185 Rome, Italy; raffaelecavalcanti@gmail.com

**Keywords:** digital denture, dental implants, guided surgery, hybrid prosthesis, immediate loading

## Abstract

Purpose: To describe a technique intended to transfer of the intermaxillary and occlusal relationships in a fully digital environment from a complete denture to an implant-supported 3D-printed hybrid prosthesis (an acrylic resin complete fixed dental prosthesis supported by implants). Methods: In edentulous cases, the physiological mandibular position should be determined before the immediate loading procedures. In some cases, the use of interim removable prostheses for a few weeks could be useful to test the new occlusion in centric relation and to verify the prosthetic project. When the correct intermaxillary relationships are achieved, it is difficult to transfer them from the provisional to the final prostheses, as impressions or scans of edentulous arches do not have reference points for intermaxillary records. This paper presents a complex case and the technique used to transfer information from a complete denture to an implant-supported prosthesis with a digital workflow. A prosthetic stent has been used to scan the edentulous mandibular arch and to record the intermaxillary relation. Results: The delivery of the hybrid implant-supported prostheses was carried out with no problems and minimal occlusal adjustments. The patient was extremely satisfied with the treatment and the situation remained stable at the 1-year follow up. Conclusions: The approach described in the present article predictably maintains prosthetic information and allows the delivery of a final implant-supported restoration with the same occlusal relationship as the one tested with the provisional diagnostic dentures.

## 1. Introduction

Immediate loading has been proposed and used to rehabilitate edentulous mandibular arches [1]. By definition, loading of implants can be considered immediate when a fixed restoration in occlusal function is delivered within 48 h after implant placement [2,3]. Immediate loading in the mandible is a well-documented and scientifically validated approach both for overdentures and fixed prosthesis; good results with lower scientific evidence can be retrieved for immediate loading in the maxilla [3]. In the last two decades, intense research, both in the field of implant surface and design and of prosthodontic materials and procedures, has led to several immediate loading protocols, in order to reduce treatment time and to reach immediacy in the delivery of implant-supported restorations [4,5,6,7,8]. However, in many cases of patients showing edentulous arches or hopeless dentitions, the prosthetic part of the treatment could be difficult because these patients very often present alterations in the mandibular position, especially when they are used with inadequate old prostheses with worn acrylic teeth or when they experienced flaring of periodontally compromised teeth with loss of vertical dimension [9]. In such cases, the physiological mandibular position should be determined before treating the edentulous arches with the immediate loading procedures, and the use of interim removable prostheses for a few weeks can be extremely useful [10]. A comprehensive approach to these complex rehabilitations has been proposed, focusing on a careful diagnostic phase carried out with removable prosthesis and on original clinical and laboratory procedures used to record and to maintain the occlusal information found in the diagnostic phase [11]. In the last years, digital technology has entered our profession, and many procedures have been deeply renovated. The development of intraoral scanners, 3D printers, and Computer-Aided Design-Computer Aided Manufacturing (CAD/CAM) machines with superior performances, as well as the introduction of improved materials, is shifting the paradigm towards a digital workflow [12,13,14]. This paper presents the evolution of the BARI technique (BARI is the capital city of the region where all the authors come from), which permits the transfer of the intermaxillary and occlusal relationships in a digital environment from the complete denture to the implant-supported 3D-printed hybrid prosthesis.

## 2. Case Description

One of the main advantages of removable prostheses is the possibility to easily modify them several times until a satisfactory esthetic and functional result is obtained. They can be used to test new intermaxillary and occlusal relationships and thereafter should be replicated with a different and more durable material as a definitive prosthesis. The BARI technique has been used to transfer the prosthetic information (vertical dimension, centric relation, frontal and lateral guidances, and esthetics) recorded in the provisional and diagnostic phase with a removable denture to a fixed implant-supported restoration. The digital BARI technique is now presented with the aid of a clinical case. The patient was 57 years old and presented edentulous maxilla and mandible and was classified as ASA 1 (according to the American Society of Anesthesiologists Physical Status Classification System) as he had no significant systemic disease, and he was not taking any medication. The patient received the maxillary and the mandibular dentures according to the Computer-Aided Design-Computer Aided Manufacturing (CAD/CAM) Denture protocol (Figure 1, Figure 2, Figure 3, Figure 4 and Figure 5) [15,16].

The CAD/CAM denture bases were produced from a pink Poly-methyl-methacrylate (PMMA) disk (Ivobase CAD Color 34V, Ivoclar Vivadent, Schaan, Liechtenstein), while teeth were manufactured from a white PMMA disk (Vivodent CAD Multi A2, Ivoclar Vivadent, Schaan, Liechtenstein) and luted together. After 18 months with the removable prostheses, the patient felt ready and asked for a more efficient restoration in the mandible. The first concern at this moment was to carefully check the previously manufactured dentures in their esthetic and functional aspects. The patient was very satisfied with the esthetic and the occlusal aspects of the dentures but asked for a more stable solution in the lower arch. Both overdenture and hybrid prostheses (an acrylic resin complete fixed dental prosthesis supported by implants) with different numbers of implants were proposed and thoroughly explained to the patient, and his preferred choice was a hybrid prosthesis supported by six mandibular implants. Guided surgery was planned for this patient as it has demonstrated to be an accurate method to reduce the probability of damage to anatomical structures and to simplify prosthetic treatment [17,18,19]. Intraoral and extraoral pictures of the patient with and without the prosthesis were taken. Duplicates of the maxillary and mandibular dentures were easily manufactured with a 3D printer (Rapid Shape P20, Rapid Shape GmbH, Heimshein, Germany) as the previous prosthesis project was already available. The resin used for this duplicate was radiotransparent (SHERAprint-model plus UV sand, Shera, Lemforde, Germany), and 10 radiopaque resin reference dots (ACRY C&B color RX, Rutinium Group, Badia Polesine, Italy) were added so as to create a radiographic stent, which was scanned with a CAD laboratory scanner (Scanner d710, Wieland Dental, Pforzheim, Germany) and STL files obtained (Figure 6).

A Cone Beam Computed Tomography was then performed with the mandibular radiological stent and a DICOM file obtained. The STL and the DICOM files were matched by superimposition on the reference points using proper software (coDiagnostiX 9.12 Dental Wings Inc, Montreal, QC, Canada) [20] (Figure 7).

As the patient was very happy with the appearance and occlusion of its removable prosthesis, we could use his previous prosthetic project and the STL files. Implants were placed in the virtual environment of coDiagnostiX software with care to position them in the correct prosthetic position according to the patient’s prosthesis. We were also able to select the intermediate abutments on the implants in the virtual environment. Sleeves for guided surgery marketed by the implant manufacturer were chosen and positioned, and a surgical guide with mucous support was designed with special care to maintain three points of occlusion with the opposing denture. Three fixation pins were also planned in order to stabilize the guide during the surgical procedures. The surgical stent was created according to the Layer Plastic Deposition (LPD) printing technology and tried in before initiating the surgical operations. The patient was asked to bite on the occlusal reference points provided on the stent so as to maintain it in the correct position, and the guide showed a satisfactory fit and stability [21] (Figure 8).

Surgery was performed under local anesthesia. Three pins were inserted at this stage in order to fix the surgical stent to the patient’s mandible (Anchor pin, Institut Straumann AG, Basel, Switzerland). The patient’s arches were maintained in occlusion during the insertion of the pins, and once the stent was fixed, the patient was allowed to disclose its arches, and the maxillary denture was removed. Microflaps were performed with microblades in order to displace the keratinized tissue, which was reduced in width as often happens in edentulous mandibles. Six implants with a macrodesign especially designed for immediate loading procedures and a surface intended to enhance osseointegration (SLActive BLX implants, Institut Straumann AG, Basel, Switzerland) were positioned following the guided surgery protocol (Figure 9).

The surgical guide was then removed, and the planned intermediate abutments (SRA Abutment, Institut Straumann AG, Basel, Switzerland) were connected to the implants (Figure 10).

The position of implant scan bodies within the dental arches with limited edentulous areas is recorded with intraoral scanning (IOS) devices and results in a virtual cast displaying the scan bodies. With the knowledge of scan body dimensions, the spatial position of each implant connected to a scan body is reconstructed. As this was an edentulous case, some problems had to be managed: few studies support the use of IOSs for impression capture on multiple implants aimed at the manufacture of extended implant-supported restorations as full arches. This limitation is determined by the acquisition methods of IOS and the difficulty of reconstructing extended surfaces [22]. In addition, obtaining accurate digital scans of the arch where there are large homogeneous areas, such as the spaces between implants in edentulous arches, is especially difficult. These difficulties are due to the absence of anatomic irregularities in the area to be scanned [23].

For this reason, an auxiliary device was built with a double function: to create the irregularities needed in the scanning area and to record the intermaxillary relationship. This special prosthetic stent in resin was designed and 3D-printed (SHERAprint-model plus UV sand, SHERA, Lemforde, Germany) from the digital project of the surgical guide, with mucosal support especially in the lower incisors area and in the region of the trigoni. It also had the same three occlusal stops and the same sleeves for the pins as the surgical stent. If compared to the surgical guide, the prosthetic stent had six holes for the scan bodies that were bigger than the sleeves holes and was very thin in the scan body regions. The increased space between the stent and the scan bodies allowed the intraoral scanner to recognize the scan bodies with the stent in place (Figure 11 and Figure 12).

A digital impression (Trios 3 intraoral scanner, 3Shape A/S, Copenhagen, Denmark) with the registration stent in place was taken, and due to the occlusal reference points, it was possible also to register the bite on the left and right sides (Figure 13, Figure 14, Figure 15 and Figure 16). The registered occlusal relationship was exactly the same as the one the patient had with the two dentures (which was for him very comfortable as discussed in the treatment-planning phase).

The scan of the maxillary prosthesis was easily taken outside the mouth.

At this stage of the treatment, the following STL files were available: one STL file for the maxillary denture, one for the mandibular implants plus the stent, and two STL files for the occlusion.

As the lower impression with implants and stent was missing the soft tissues, those were obtained in the virtual environment by superimposition of the lower denture scan and the prosthetic stent.

All the information previously recorded regarding interarch relationships could be used for the implant-supported rehabilitation. The long phase with the removable dentures was conducted with care and with a great effort to find the correct interarch relationships. These data should be kept, and with this technique, no information was lost. All the needed information was present in the digital environment at this stage: the scan of the mandibular arch with the implants and the stent in place, the scan of the opposing maxillary denture, the file of the previous mandibular denture, and the bite registration with the intermaxillary relationship taken with the prosthetic stent. All these files could be superimposed because of consistent reference points.

It was possible to design a hybrid prosthesis using the libraries of the variobase titanium copying matching with the SRA abutments. The hybrid prosthesis was 3D printed (NEXTDENT resin C&B, Nextdent B.V., Soesterberg, the Netherlands) and tried in on 3D-printed models (SHERAPRINT Model Plus UV GREY Resin, SHERA, Lemforde, Germany). It was then stained (Optiglaze, GC Europe, Leuven, Belgium) and refined, and afterward, the variobase copings (Institut Straumann AG, Basel, Switzerland) were luted using the digital models as reference (Figure 17 and Figure 18).

After polishing, the manufactured prosthesis was delivered to the patient a few hours after the surgery in which the screws were tightened at 15 N/mm (Figure 19, Figure 20 and Figure 21). The care taken in the procedures of information transfer allowed the avoidance of major occlusal adjustment at the moment of delivery and the maintenance of the esthetic and functional conditions already extensively tested for several weeks in the patient’s mouth. The provisional diagnostic phase with the removable dentures was conducted with care and with great effort to find the correct interarch relationships. All the information previously recorded regarding interarch relationships was not lost and could be used for the implant-supported rehabilitation.

## 3. Results

The delivery of the hybrid implant-supported prostheses was carried out with no problems and minimal occlusal adjustments, meaning that the technique allowed for a precise transfer of prosthetic information. The patient entered a maintenance program with regular visits for professional oral hygiene and checkups three times a year. At the one-year follow up, the situation remained stable: implants were healthy with no signs of tissue inflammation, and the prostheses were functioning correctly. The patient was extremely satisfied with the treatment, and the prognosis of the rehabilitation is very good, as the literature shows excellent long-term results for the rehabilitation delivered to the patient.

## 4. Discussion

The prosthetic stent used in the present paper had a double function. First, the stent was useful for taking an accurate intraoral scan, as it is well documented that edentulous arches represent critical clinical situations for intraoral scanning. The use of an auxiliary stent in order to increase the accuracy of intraoral scans has been described and tested in previous papers [24,25].

The device used in the present paper accomplished a second very important function: the registration of the occlusal relationship. When an edentulous arch is restored, the prostheses should be designed with occlusion in centric relation, the maxillomandibular relationship in which the condyles articulate with the thinnest avascular portion of their respective disks with the complex in the anterior-superior position against the shapes of the articular eminencies. This position and the correct vertical dimension are difficult to find, as edentulous patients very often present with lost occlusal references. The process of recording and testing the correct intermaxillary relations, vertical dimension, phonetics, and esthetics could be very challenging and could take a lot of time [11]. The same issue is faced in patients with hopeless dentition, who very often present with pathologic migration of the periodontally compromised teeth and with posterior bite collapse [26]. Provisional prostheses (either fixed or removable) have at least two functions: they obviously provide acceptable function and esthetics to patients until final prosthesis are ready, and they can be also used as a diagnostic tool, as they can be modified or changed many times until a satisfactory esthetic and functional result is achieved.

As the provisional phase takes time and effort, once a satisfactory result is obtained, the final prostheses should be copies of the provisional ones made of more esthetic and more durable materials. Removable dentures are frequently used by the authors in the initial diagnostic phase (diagnostic dentures), providing the patient with an adequate occlusal scheme which can recondition the neuromuscular masticatory apparatus. These dentures allow the mandible to physiologically find its natural position and can be modified several times until the correct mandibular position, pleasant aesthetics, and comfortable phonetics are obtained.

One of the difficulties in the dental routine is to step from a satisfactory provisional to the final restoration. In the analogic workflow, new impressions are taken for the final prosthesis, and they should be mounted in the articulator. In cases of fixed restorations, the full arch provisional is used to record the face bow and to mount the casts in the articulator. A technique to digitally transfer the prosthetic information (vertical dimension, centric relation, frontal and lateral guidances, and esthetics) in cases treated with fixed provisional prostheses has been proposed [27]. The registration of interarch relationships to deliver a 3D-printed bite for TMJ patients has also been described [28].

The difficulty of patients with edentulous arches is related to the absence of fixed reference points. A BARI technique was described to transfer the jaws relationship when the provisional and diagnostic phase is performed with a removable denture. In the BARI technique, a prosthetic stent was used to transfer implant positions: the implant transfer copings were blocked with resin to the stent and the stent-transfer analogs complex was repositioned on the excavated master model [11]. The process of luting the transfer to the prosthetic stent and the phase of master model excavation and analogue reposition in stone are time consuming at the dental chair and in the laboratory, as potential errors in the complex procedure could be incorporated in any step of the workflow. The approach described in the present article is an evolution of the concept in a fully digital workflow. Using this approach, all the information can be maintained in a predictable manner, and a final implant-supported restoration with the same occlusion as the one tested with the provisional diagnostic dentures can be delivered to the patient. The possibility of dealing with a fully digital protocol makes this process easier and faster: only the intraoral scan with the stent and the registration of the bite are needed from the dental office, and all the superimpositions are done in the laboratory and could be done very easily, as the prosthetic stent is made from the patient’s denture STL file and shares a lot of reference points.

Another advantage of the digital environment is its flexibility to produce a CAD/CAM or a 3D-printed hybrid prosthesis. The diffusion of 3D printers in the dental labs and offices is constantly growing, and the possibility to rapidly produce a prosthesis at reasonable costs is really promising.

## 5. Conclusions

Intraoral scans of edentulous arches may be challenging, mainly for two reasons: it is difficult for the IOS devices to scan large edentulous areas and it is very complex to record the correct inter-arch relationship.

The present paper describes a dental technique intended to solve both problems with the aid of a prosthetic stent. The proposed approach has been used to successfully rehabilitate the case described in the manuscript. Further studies with larger samples are needed in order to validate the presented workflow.

## Figures and Tables

**Figure 1 ijerph-16-05160-f001:**
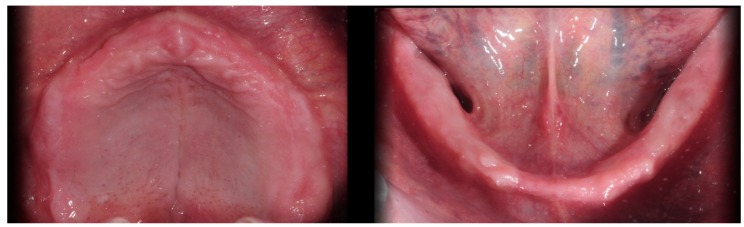
Occlusal views of the edentulous arches of the patient at the beginning of the treatment.

**Figure 2 ijerph-16-05160-f002:**
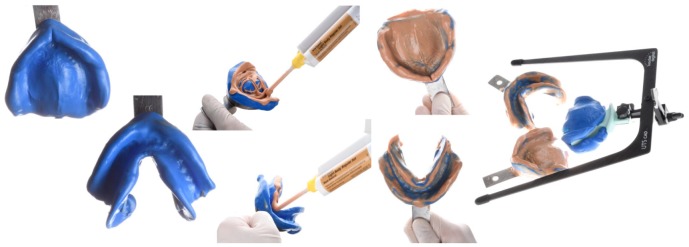
Preliminary impressions of the edentulous arches and the centric tray with the Universal Transfer Bow Computer-Aided Design (CAD) registration.

**Figure 3 ijerph-16-05160-f003:**
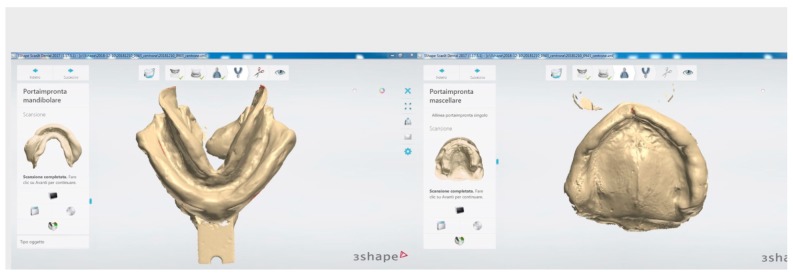
Scans of impressions.

**Figure 4 ijerph-16-05160-f004:**
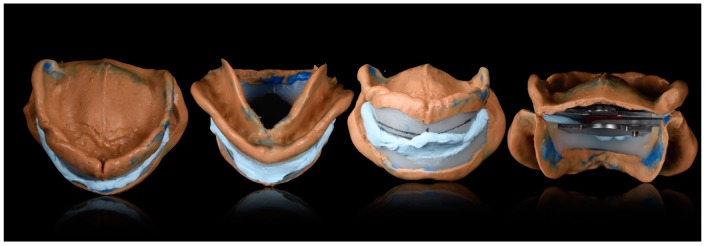
Different views of functional impressions with base plates and intermaxillary recording ready to be scanned and imported in the digital environment.

**Figure 5 ijerph-16-05160-f005:**
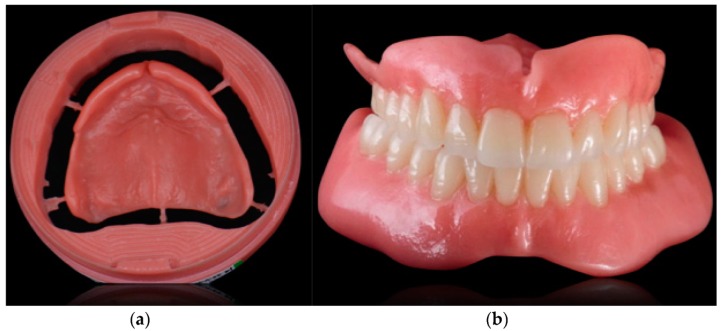
(**a**) CAD/CAM denture base immediately after its milling (**b**) final dentures after the luting of teeth and the polishing and refining phase.

**Figure 6 ijerph-16-05160-f006:**
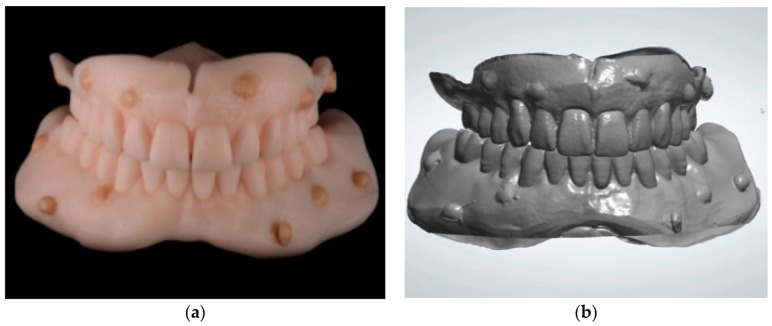
(**a**) Duplicates of the maxillary and mandibular dentures with radiopaque resin dots; (**b**) scans of the denture duplicates.

**Figure 7 ijerph-16-05160-f007:**
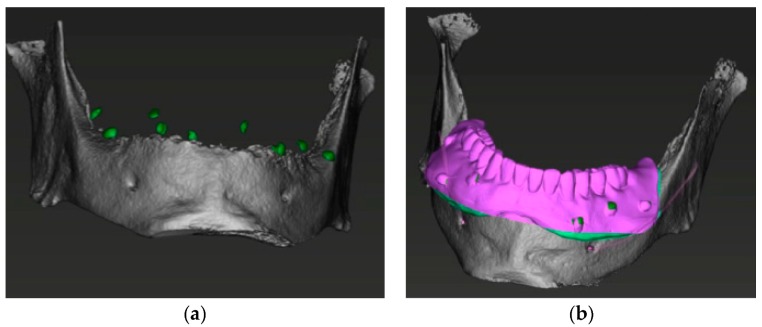
(**a**) 3D mandibular reconstruction from DICOM file; (**b**) superimposition of the mandible and the prosthesis made on the 10 reference dots.

**Figure 8 ijerph-16-05160-f008:**
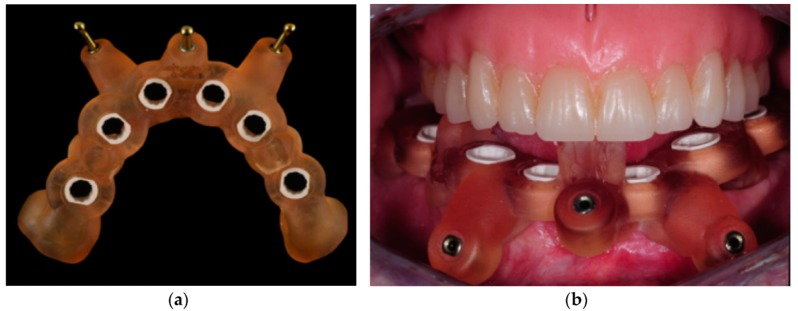
(**a**) Surgical stent ready after its production process; (**b**) Surgical stent kept in place before insertion of the pins by the maxillary prosthesis contacting on three reference points.

**Figure 9 ijerph-16-05160-f009:**
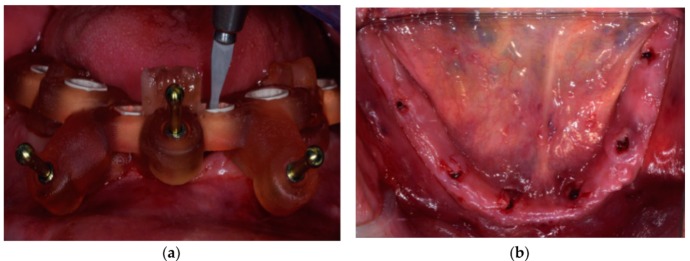
(**a**) Surgical stent in place after the insertion of anchor pins. Microflaps were performed with microblades in order to preserve keratinized tissue; (**b**) occlusal view immediately after implant placement and after the removal of the surgical stent.

**Figure 10 ijerph-16-05160-f010:**
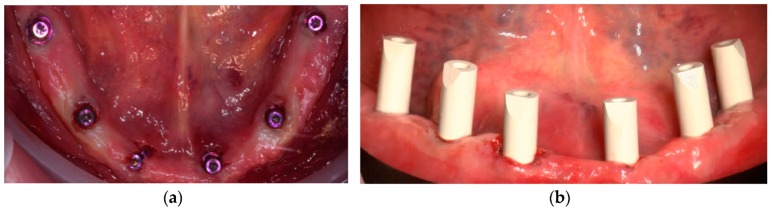
(**a**) Occlusal view immediately after the connection of SRA abutments on implants; (**b**) frontal view of scan bodies connected to SRA abutments.

**Figure 11 ijerph-16-05160-f011:**
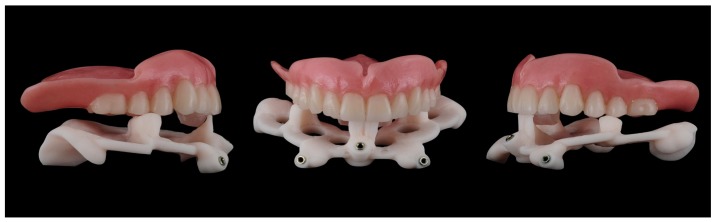
Frontal and lateral views of the prosthetic stent.

**Figure 12 ijerph-16-05160-f012:**
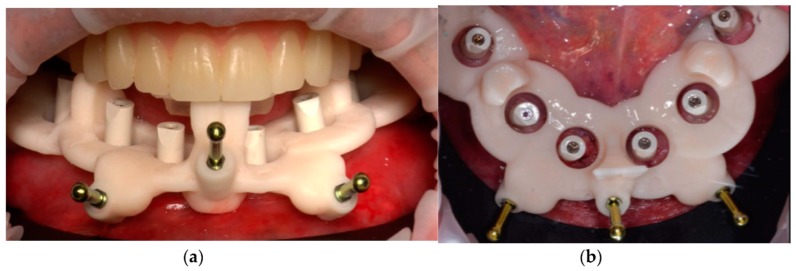
Frontal and occlusal views of the prosthetic stent in place. The prosthetic stent was kept in situ with the maxillary prosthesis and then fixed with the same anchor pins used to fix the surgical stent.

**Figure 13 ijerph-16-05160-f013:**
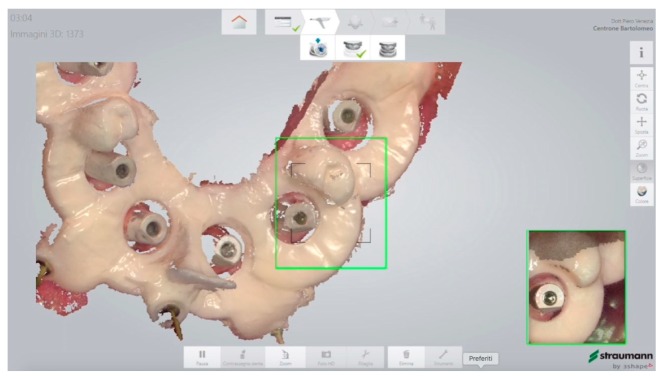
Scan of the mandibular arch with the prosthetic stent in situ.

**Figure 14 ijerph-16-05160-f014:**
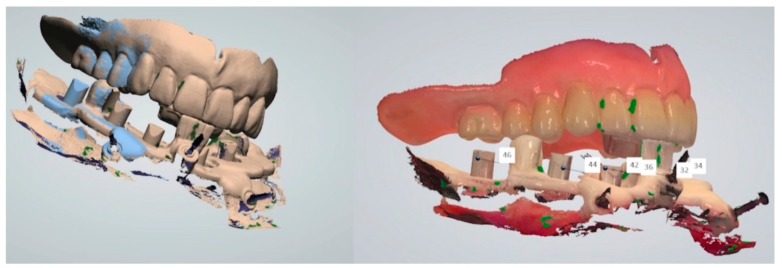
Bite registration of maxillary prosthesis with the prosthetic stent and scan bodies.

**Figure 15 ijerph-16-05160-f015:**
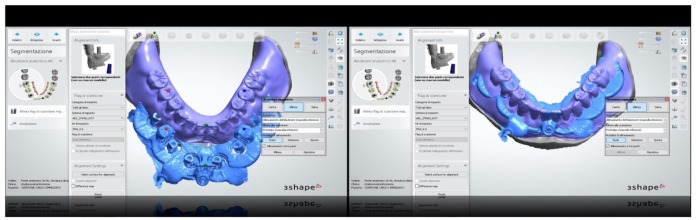
Superimposition of the original prosthetic project and of the prosthetic stent with scan bodies.

**Figure 16 ijerph-16-05160-f016:**
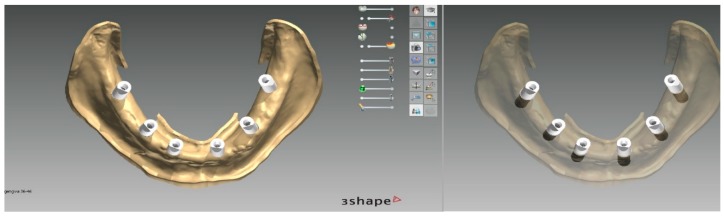
Mandibular model with scan bodies at SRA abutment level in situ (obtained by superimposition of the lower arch intraoral scan with the prosthetic stent in situ and the scan of the patient’s mandibular prosthesis to provide soft tissues information).

**Figure 17 ijerph-16-05160-f017:**
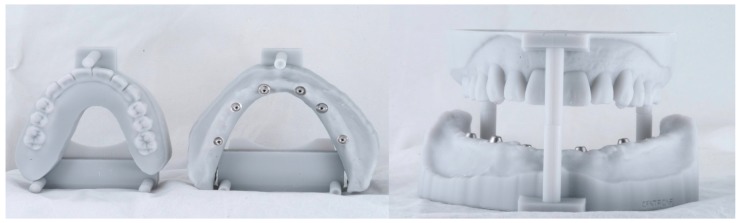
3D-printed models in the correct occlusal relations.

**Figure 18 ijerph-16-05160-f018:**
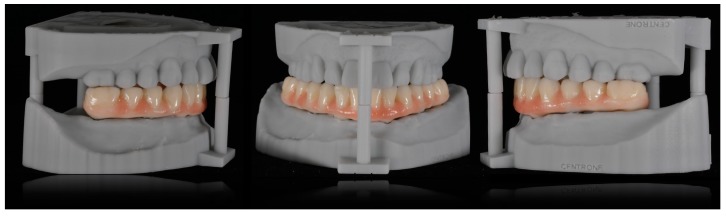
Frontal and lateral views of the hybrid prosthesis produced on 3D printed models.

**Figure 19 ijerph-16-05160-f019:**
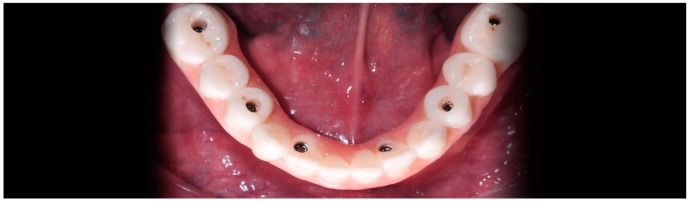
Hybrid prosthesis delivered to the patient a few hours after implant placement.

**Figure 20 ijerph-16-05160-f020:**
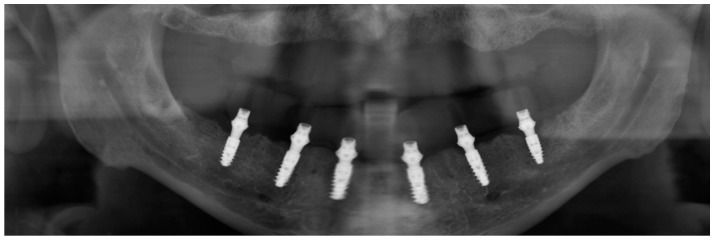
Panoramic control X-ray.

**Figure 21 ijerph-16-05160-f021:**
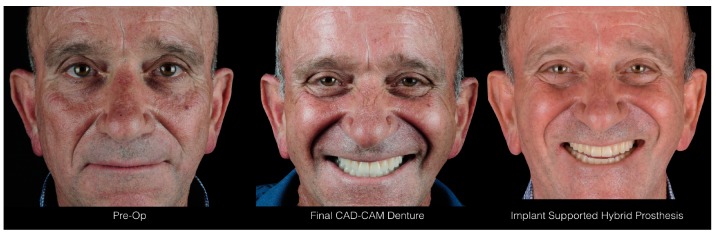
Patient’s photos before the treatment, after the delivery of the dentures, and after the delivery of the implant-supported hybrid prosthesis.

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
