# Peer review of "Full Digital Workflow for the Treatment of an Edentulous Patient with Guided Surgery, Immediate Loading and 3D-Printed Hybrid Prosthesis: The BARI Technique 2.0. A Case Report"

_ijerph, 2019, doi:10.3390/ijerph16245160_

Round 1

Reviewer 1 Report

this is a brillant case report, i congratulate with the authors for the quality of this paper.

there are only a few minor typos/ errors in the english form, i suggest the authors to check but the paper is really of high quality and the pictures, beautiful.

when you use abbreviations, please specify what they stand for at least at first use!

i would love if the authors could expand their discussion section focusing on the differences between the analog approach and the digital approach, particularly with the BARI technique, giving advantages and limits of both the approaches, this is important. some concepts that are challenging for the dentists, such as the vertical dimension, and the centric relation, should be better developed in the section.

the declaration session must be completed, i suggest the authors to carefully read the instructions to the authors.

Author Response

Thank you for the nice review.

We've tried to improve English language and to correct errors.

extended names were used instead of abbreviation at the first use as requested by the referee. 

Centric relation concept has been explained in the discussion

Analogic-digital comparison has been performed in the discussion

Reviewer 2 Report

excellent case report that deserves publication in the IJERPH. the problem is mainly related to the quality of the english language that should be improved. i suggest to ask the help of a proficient english speaker.

just a few comments to further improve the quality:

in the abstract, purpose should not be in bold characters in the abstract, what do you mean for "hybrid" prosthesis? this should be elucidated because the readers may not be aware of it in the abstract, methods, the sentence " In edentulous cases the physiological mandibular position should be determined before
treating them with the immediate loading procedures" is not in proper english. what do you mean here for "them"?  in the abstract, you really do not provide sufficient information about the methodology of your case report and you do not describe in sufficient details your procedures, you need therefore to expand this section and give more details in the intro, i would like you to expand and go deeper when you explain why it is difficult to register the correct mandibular position, in the edentulous patient or in patients with complete dentures of insufficient quality. can you explore the concept of centric relation, vertical dimension, giving more details on the techniques that are currently available with conventional (analog) workflows? and what is the potential benefit of the digital workflow? what is the impact? can we simplify these procedures?  in the intro, at the end, "CAD/CAM machines" should read "CAD/CAM systems" and you should first specify, what CAD/CAM actually stands for, i.e. computer-assisted-design/ computer-assisted-manufacturing (CAD/CAM)  what do you mean for "BARI" technique? what this acronym means? please explain in the case description/ methods, page 2, please specify what ASA is. "The patient received two maxillary and the mandibular dentures according to the CAD CAM Denture protocol [15,16]" englishg sentence not clear here. two maxillae? in the methods, Fig. 2, what do you mean for "UTS"? in the methods, page 5, " Implants have been placed in the virtual environment of co-diagnostix software with care to position them in the correct prosthetic position according to the patient’s
prosthesis". Co-Diagnostic is a proprietary software from Dentalwings/ Straumann group and this should be disclosed accordingly Fig. 8, why the guide is yellow? how did you perform "microflaps" to preserve keratinized tissue locally, with such a big guide that looks like an occlusal bite? isn't it very challenging? can you disclose the brand of the intraoral scanner you used for taking the impression of the scanbodies/ scanabutments? i think, Trios from 3Shape. you should add this info in the body of your manuscript Fig. 21, i think you should cover the eyes of your patient and obviuosly, you should disclose in the paper that you have his written consent, to use his personal pictures. This should be disclosed also in the Declarations section, at the end of the manuscript Results: no references should be included in the Results section, please erase cit. n°3 from there Discussion is really too short. i would appreciate if you could explain the concepts of mandibular position, centric relation, vertical dimension etc in order to speak about the problems with the conventional workflow, how generally dentists face these problems analogically, and then the potential of the present digital approach that looks really promising. after at least 3-4 paragraphs dedicated to this, you should report in details on your previous paper on the BARI technique, describing in details what you did. Then, you should report on the results and the advantages with the digital BARI technique. please do not be too concise, the readers of IJERPH would love it if you expand your discussion.  please give 1-2 sentences on the limitations of this study - just a case report more studies are needed etc no Conclusions here?  the Declaration section must be completed with the Consent for publication- you need a Consent signed from the Patient if you want to publish his pictures Reference list is appropriate but could be slightly expanded figures are really of high quality

Author Response

Dear Referee,

thank you for your nice review. we've tried to fulfill all your requests.

"purpose" is not anymore in bold characters in the abstract

 "hybrid" prosthesis has been elucidated as requested

methods, the sentence " In edentulous cases the physiological mandibular position should be determined before
treating them with the immediate loading procedures" is not in proper English. what do you mean here for "them"? 

modified as requested

in the abstract, you really do not provide sufficient information about the methodology of your case report and you do not describe in sufficient details your procedures, you need therefore to expand this section and give more details in the intro, i would like you to expand and go deeper when you explain why it is difficult to register the correct mandibular position, in the edentulous patient or in patients with complete dentures of insufficient quality.

the abstract has been elongated as requested

can you explore the concept of centric relation, vertical dimension, giving more details on the techniques that are currently available with conventional (analog) workflows? and what is the potential benefit of the digital workflow? what is the impact? can we simplify these procedures? 

DONE in the discussion

in the intro, at the end, "CAD/CAM machines" should read "CAD/CAM systems" and you should first specify, what CAD/CAM actually stands for, i.e. computer-assisted-design/ computer-assisted-manufacturing (CAD/CAM) 

DONE

what do you mean for "BARI" technique? what this acronym means?

BARI is the city where the authors come from (it has been now explained in the text)

please explain in the case description/ methods, page 2, please specify what ASA is.

ASA 1 according to the American Society of Anesthesiologists Physical Status Classification System). This is now added to the manuscript 

"The patient received two maxillary and the mandibular dentures according to the CAD CAM Denture protocol [15,16]" englishg sentence not clear here. two maxillae?

correct sentence is: the patient received the maxillary and the mandibular dentures according to the CAD CAM Denture protocol [15,16]

in the methods, Fig. 2, what do you mean for "UTS"?

we've added full name added to the legend

n the methods, page 5, " Implants have been placed in the virtual environment of co-diagnostix software with care to position them in the correct prosthetic position according to the patient’s
prosthesis". Co-Diagnostic is a proprietary software from Dentalwings/ Straumann group and this should be disclosed accordingly 

It was disclosed in the previous sentence.

Fig. 8, why the guide is yellow? how did you perform "microflaps" to preserve keratinized tissue locally, with such a big guide that looks like an occlusal bite? isn't it very challenging?

yes, it is challenging but it is important to displace the KT and not to cut it with burs. We did it with micro blades as depicted in fig. 8

the yellow color is the color of this resin especially intended for surgical guides. it is a resin which can be sterilized in the autoclave unit.

can you disclose the brand of the intraoral scanner you used for taking the impression of the scanbodies/ scanabutments? i think, Trios from 3Shape. you should add this info in the body of your manuscript

DONE

Fig. 21, i think you should cover the eyes of your patient and obviuosly, you should disclose in the paper that you have his written consent, to use his personal pictures. This should be disclosed also in the Declarations section, at the end of the manuscript

DONE. we have consent to use uncovered pictures.

Results: no references should be included in the Results section, please erase cit. n°3 from there DONE

Discussion is really too short. i would appreciate if you could explain the concepts of mandibular position, centric relation, vertical dimension etc in order to speak about the problems with the conventional workflow, how generally dentists face these problems analogically, and then the potential of the present digital approach that looks really promising. after at least 3-4 paragraphs dedicated to this, you should report in details on your previous paper on the BARI technique, describing in details what you did. Then, you should report on the results and the advantages with the digital BARI technique. please do not be too concise, the readers of IJERPH would love it if you expand your discussion. 

DONE

please give 1-2 sentences on the limitations of this study - just a case report more studies are needed etc no Conclusions here? 

DONE

the Declaration section must be completed with the Consent for publication- you need a Consent signed from the Patient if you want to publish his pictures DONE

Reference list is appropriate but could be slightly expanded DONE

figures are really of high quality Thank you!